# Biliary Atresia in Adolescence and Adult Life: Medical, Surgical and Psychological Aspects

**DOI:** 10.3390/jcm12041594

**Published:** 2023-02-17

**Authors:** Deirdre Kelly, Marianne Samyn, Kathleen B. Schwarz

**Affiliations:** 1Liver Unit, Birmingham Women’s & Children’s NHS Hospital, University of Birmingham, Birmingham B15 2TT, UK; 2Paediatric Liver, Gastroenterology and Nutrition Unit, King’s College Hospital NHS Foundation Trust, London WC2R 2LS, UK; 3Pediatric Liver Center, Johns Hopkins University School of Medicine, Baltimore, MD 21287, USA; 4Pediatric Liver Center, UCSD School of Medicine/Rady Children’s Hospital, San Diego, CA 92123, USA

**Keywords:** liver, biliary atresia, transition, adolescence, adherence, readiness, professionals, parents

## Abstract

Prior to 1955, when Morio Kasai first performed the hepatic portoenterostomy procedure which now bears his name, Biliary atresia (BA) was a uniformly fatal disease. Both the Kasai procedure and liver transplantation have markedly improved the outlook for infants with this condition. Although long-term survival with native liver occurs in the minority, survival rates post liver transplantation are high. Most young people born with BA will now survive into adulthood but their ongoing requirements for health care will necessitate their transition from a family-centred paediatric service to a patient-centred adult service. Despite a rapid growth in transition services over recent years and progress in transitional care, transition from paediatric to adult services is still a risk for poor clinical and psychosocial outcomes and increased health care costs. Adult hepatologists should be aware of the clinical management and complications of biliary atresia and the long-term consequences of liver transplantation in childhood. Survivors of childhood illness require a different approach to that for young adults presenting after 18 years of age with careful consideration of their emotional, social, and sexual health. They need to understand the risks of non-adherence, both for clinic appointments and medication, as well as the implications for graft loss. Developing adequate transitional care for these young people is based on effective collaboration at the paediatric–adult interface and is a major challenge for paediatric and adult providers alike in the 21st century. This entails education for patients and adult physicians in order to familiarise them with the long-term complications, in particular for those surviving with their native liver and the timing of consideration of liver transplantation if required. This article focusses on the outcome for children with biliary atresia who survive into adolescence and adult life with considerations on their current management and prognosis.

## 1. Introduction

The Kasai hepatic portoenterostomy, which was first performed in 1955 [1], markedly improved the outlook of children born with biliary atresia (BA). While there are a number of studies to determine short-term outcome following the Kasai, data have emerged more slowly regarding long-term outcomes. The purpose of this manuscript was to review overall actuarial survival data up to 30 years post-Kasai, the health status of survivors both with native liver and post-transplant, and predictors of outcome and strategies to improve it. To achieve this review, the first 100 papers listed in Pub Med under the search term “biliary atresia long term survival” were inspected and select ones were summarised, with emphasis on the most detailed papers published within the last five years.

## 2. Overall Survival Data up to 30 Years

One of the largest and most recent series published is that of Fanna et al. [2] who summarised results on the 1428 patients with BA managed in France from 1986 to 2015. In addition to their own series, the authors reported a comprehensive analysis of 11 BA registries in Europe, America and Asia. (Table 1). Survival with native liver after the Kasai operation at 10 years was 26 to 73% (average of means 44%); at 20 years after the Kasai in the 3 registries reporting was 24–28%, and at 30 years after the Kasai was 22% for France and 49% for Japan. Survival post liver transplant (LT) was 86% at 5 years to 79% at 30 years. In total, 10.2% died without LT.

## 3. Health Status of Survivors

### 3.1. Chronic Liver Disease

Kelay and Davenport [3] described the long-term health status of survivors with native liver. Only 11% had no signs of liver disease. Manifestations of chronic liver disease included cirrhosis in most after age 20 years, portal hypertension/varices, and recurrent cholangitis. Malignant transformation was rare but did occur: hepatocellular carcinoma developed in 0.8% and there were rare cases of hepatoblastoma and cholangiocarcinoma. Interestingly growth and development were usually normal.

### 3.2. Health Related Quality of Life (HRQOL)

Rodijk et al. [4] reviewed the literature on HRQOL in children with biliary atresia and noted that, in general scores in children with BA were lower than in healthy peers. They also performed their own study in Dutch BA patients (*n* = 38; age 10 ± 3 years), the parent-proxy physical score was significantly lower compared with healthy controls. It was also lower than in children with a variety of other medical conditions. Psychosocial HRQOL was lower than in healthy peers and largely comparable to children with other chronic conditions. Parent-proxy physical HRQOL was adversely related to adverse medical events in the past year, special education, and motor impairments; psychosocial HRQOL was adversely related to behavioural problems. Liang et al. [5] also observed that having other medical conditions impacted negatively on the HRQOL; an additional factor that impacted the HRQOL was the parents’ knowledge of LT [5]. In general, HRQOL of parents of children with BA was adversely affected [6]; Wong et al. [7] performed similar analyses and noted that HRQOL scores did not differ in general between those with a native liver and those post LT.

### 3.3. Neurodevelopmental Outcome

The Childhood Liver Disease Research and Education Network (ChiLDREN) performed comprehensive neurodevelopmental testing on children with BA and their native liver, ages 3–12 [8]. As shown in Figure 1 [8], iIn general average IQ scores were above the expected. (as indicated by light gray). However, Ruuska et al. [9] found the opposite Rodiijk et al. [10] studied 46 Dutch children ages 6–12 years with BA in a group some of whom had had LT and some who were with native liver. Motor delays were particularly prominent [10]. No longer-term studies of neurodevelopmental testing in children with BA were identified.

### 3.4. Predictors of Outcome

One of the major determinants of long-term outcome is the age at Kasai (Figure 2). Survival up to 30 years was 38, 27, 22, and 19% in patients operated on at 1 month, 2, 3 and later [2]. The anatomic pattern of the biliary remnant and the presence or absence of Biliary Atresia Splenic Malformation (BASM) were also important determinants [2,3]. The data regarding the impact of these factors on long-term survival up to 30 years are shown in Table 2. As noted by Kelay [3], the etiology of BA also influenced outcome with worse outcome for cat eye syndrome (chromosome 22 aneuploidy), and better outcome for cystic BA. The type of anatomy influenced the type of operation and had an impact on 20-year survival: hepatic portocholecystostomy 35% cystojejunostomy 40%, Type IIIa 19%.

A very recent report by van Wessel et al. [11] noted that the gut microbiota composition of BA patients pre-Kasai associated with outcome at 6 months. No data are yet available re the effect on long-term outcome. Interestingly enough, cytokines were not predictive of outcome Madadi-Sanjaniet al. [12]. Harumatsu et al. [13] recently reported that microvascular proliferation of the portal vein branches at the time of Kasai was associated with better outcome; Sasaki et al. [14] made similar observations. Finally, Johansson et al. [15] recently reported that reduction of hepatic FGF 19 at the time of Kasai predicted better outcome.

Wang et al. [16] noted a strong difference between the estimated 5-year native liver survival (NLS) rates of their successful Kasai group (defined as clearance of jaundice at 4 weeks post Kasai) and failed Kasai group: NLS rates at 5 years post Kasai were 90.1% vs. 10.7% for successful vs. failed Kasai (*p* = 0.000). Shneider et al. [17] noted similar findings for total serum bilirubin 3 months post-Kasai in the ChiLDReN BA cohort. Additionally, utilising data from the ChiLDReN study, Venkat et al. [18] found that long-term survival up to age 14 years could be modelled using platelet count, GGT, and predicted SE-free survival at age 7 years. (Figure 3).

## 4. Adolescence and Transition

### 4.1. Adolescence

Adolescence is a challenging time for all young people and their families, as they progress through multiple physical, developmental and cultural changes in parallel [19]. It is a time of increasing independence and autonomy, with a belief in one’s own immortality, struggles with peer pressure, a focus on body image, frequently undiagnosed mental health problems.

In contrast to biological and cognitive changes, the psychosocial changes of adolescence may be culturally determined and include the social “tasks” of adolescence such as establishing relationships, achieving independence from parents and establishing financial (i.e., vocational) independence [20,21]

Conversely, in the search for identity and independence, immature abstract thinking common in adolescence may make medical management difficult through poor adherence to medical regimens and “risky” health behaviours [22]. 

The additional burden of biliary atresia requiring regular hospital visits and medication can make this period even more difficult for young people to navigate, particularly when they move from family-centred paediatric care to adult services [23].

### 4.2. Transition

Transition is an active process that focusses on the medical, psychosocial and educational/vocational needs of adolescents as they move from child to adult-centred care [24,25].

The aims of transition are [26,27] to provide high quality, coordinated health care which is patient-centred, age and developmentally appropriate, flexible, and comprehensive; to encourage skills in communication, decision-making, self-care and self-advocacy; to develop a sense of control and independence in health care [28] and to maximise life-long physical and psychosocial health.

Transitional care acknowledges the reciprocal influences of adolescent development, the underlying chronic illness and/or the effects of transplantation. Patients may have been affected by their chronic liver disease with malnutrition or delayed puberty. Cognitive development may have been affected by their disease or by drug side effects (steroids and calcineurin inhibitors), pain, fatigue and repeated hospitalisation. The stage of cognitive development is important in planning health and disease education for such young people as well as their involvement in decision-making and self-care. 

The combination of chronic liver disease/post-transplant and adolescent development (physical, cognitive and psychosocial) are important considerations for the multidisciplinary teams. Both paediatricians and adult physicians should monitor growth and pubertal development especially if there has been growth retardation due to the biliary atresia. Some of these patients may have a learning disability and health providers need to be skilled at both age and development appropriate management for this age-group.

Paediatric care focusses on the relationship with parent, professional and child, and needs to change to an adult relationship of patient and professional. As they mature, young people progress from using emotional strategies (such as wishful thinking or resignation) to problem-solving strategies which may be delayed by their chronic illness [24,25,26,27].

### 4.3. The Transitional Care Programme

Transitional care should start at around 12 years of age to promote resilience and self-determination in the young person and their families. There are no good measures of transition readiness and as transition is a process, there is no agreed age for the end of transition or the timing of transfer to adult care so it should be individualised for each patient. Discussion and preparation for transition needs should be at a time of good hepatic/graft function and transfer should not be implemented during an acute illness or rejection or graft failure [29].

Although the focus should be on the young person, family relationships and stability are important factors for resilience. Lack of parental support at this time has been associated with negative outcomes, e.g., greater non-adherence to medication.

As the adolescent negotiates the tasks and transitions detailed above, his/her parent are also managing the transition from being a parent of a dependent child to a parent of an independent young adult and need appropriate support [30,31].

Ideally, there should be an identified health professional for each young person, who can support the process in the paediatric and adult units and the primary health care team. Equally essential are other key players such as teachers, (including careers and vocational rehabilitation), social services and voluntary organizations who provide dedicated youth 

Transition may take place in a variety of ways: via a transition clinic from paediatric to an adult care or a specific adolescent clinic or a young adult clinic before moving into adult care, but it is important that the Transition team, both adult and paediatric, work together.

Absolutely key to the success of the process is an interested and capable adult service, willing to continue the transitional process in adult care. Establishing a local network of interested and committed professionals is vital for the success of any transitional care pro- gramme [32,33].

### 4.4. Key Transitional Care Issues

#### 4.4.1. Disease Education

Most young people with biliary atresia with or without a LT will have had their initial disease education directed to the parents and so need to be educated about their disease/post-transplant state. The use of age appropriate literature and a skilled play worker or teacher are invaluable to help young people to understand and accept difficult issues such as having a dead person’s organ, or facing a life-time of medical monitoring and medication [34].

In addition, young people need to be aware of the signs and symptoms requiring urgent medical attention and how to access to medical care, which may be daunting for them in an adult unit. Information regarding drugs should include the importance of adherence and side effect profile and monitoring but also rationale and benefits [35].

#### 4.4.2. Generic Health Education and Sexual Health

Generic health issues need consideration especially as many adult health-promoting behaviours become established during adolescence. Adolescents with chronic illness report more age- related concerns than their healthy peers: acne, alcohol and drug, periods, headaches, anxiety, contraception, insomnia, worry about height and weight and sexual health. 

Greater levels of exercise are associated with well-being and long-term functioning in patients with chronic conditions [36] and are to be encouraged when feasible, particularly in view of the concerns of the metabolic syndrome post-transplant related to obesity and inactivity [37,38].

All young people should understand the implications of their disease/post-transplant state, e.g., immunosuppression and other treatments, on their sexual and reproductive health as they may be sexually active [35,39].

It is important to provide information about contraception. Although barrier methods are the safest, there is a high failure rate. It is safe for girls to take a low dose oestrogen/progesterone contraceptive pill or to take the ‘morning after pill’ [40]. Chronic liver disease may have delayed puberty but this will begin after a successful transplant. Most immunosuppressive drugs are not teratogenic or affect fertility apart from mycophenolate mofetil which should be avoided in adolescent girls. Many girls develop menorrhagia post-transplant and specific advice from a gynaecologist trained in managing contraception and pregnancy in patients with chronic liver disease or immunosuppression is useful.

#### 4.4.3. Substance Misuse

Cigarette smoking should be discouraged as there may be an increased risk of lung cancer post-transplant. Substance misuse has been reported in young people who are non-adherent with medication and hence an important aspect of history taking. It is important to highlight the importance of sensible behaviour, e.g., LT recipients may drink alcohol with their peers in moderation [41].

#### 4.4.4. Self-Advocacy and Psychosocial Issues

Self-advocacy skills are a key goal of transition, such as being independent from parents, having a full understanding of their illness, involvement in decision-making, self-medication, adherence, etc. The young person should begin to develop these skills within the transition clinic and become more confident in decision-making, managing their own health by developing communication skills, independent living skills, accessing the health and educational services to prepare them for adult life [42].

Seeing the young person independently from the parent helps provide the privacy for such discussions of generic health concerns such as sexual health, alcohol use. 

Coping with teasing and/or bullying as well as disclosure issues are important issues to address with the young person during adolescence and transition. Transplant recipients may be particularly vulnerable because of their altered appearance from disease or medication, or because of time lost from school. Exploring and developing coping strategies for disclosure with a health professional can help the young person gain in confidence. Peer support may be a useful means of promoting well-being for such young people in addition to psychological support [43].

Despite the many potential problems, many long-term survivors with biliary atresia report satisfactory completion of education and high levels of employment [44]. 

#### 4.4.5. Non-Adherence

Non-adherence is a key risk factor for maintaining good disease control and is a significant cause of post-transplant acute rejection, graft loss and death [45]. It remains one of the most challenging aspects of managing care in the adolescent and young adult population. It may be part of the spectrum of risk-taking behaviour or associated with other risk-taking behaviours such as alcohol and drug use. It may be intentional or non-intentional and include mean missing a dose, taking doses irregularly, changing the prescribed dose, taking a medication holiday or stopping medications altogether [46].

Management is complex but should not be judgmental as a lack of sensitivity or awareness of adolescent/young adult issues can be particularly damaging and may result in increased non-adherence and disengagement with health care services. Interventions should be tailored to the individual patient and use a combination of approaches such as wireless enabled pill-boxes, customised text reminders, provision of electronic feedback and individual behavioural sessions involving goal setting and motivational interviewing [47].

## 5. Medical and Surgical Management in Adolescence and Adult Life 

Young adults with BA surviving with their native liver are likely to develop complications in adolescence and adult life. Reports vary from 60% of patients with BA surviving long term without LT [48] to nearly two-thirds of young adults with BA developing complications requiring LT [49]. The commonest complications were cholangitis (100%), portal hypertension (80%) and variceal bleeding (45%). In contrast to adults with cirrhosis, hepatocellular carcinoma was rare (1.3%). Synthetic liver failure, recurrent cholangitis and complications of portal hypertension were the main indications for Liver transplantation [50,51].

Cholangitis is a common complication post Kasai porto-enterostomy, seen in over 50% of children in first 2 years after surgery. Less information on its prevalence in adolescence or adulthood is available. In our series of 89 young people with BA surviving with their native liver after the age of 16 years, 10% developed at least one episode of cholangitis between the age of 12 and 16 years and this was found to be associated with a 4-fold risk of requiring liver transplantation during adulthood [51] The symptoms can be non-specific and liver function tests not helpful as 90% of BA patients will have some deranged liver function tests. Cholangitis should be suspected when presenting with right upper quadrant pain, general malaise and nausea. Imaging can demonstrate the presence of dilated biliary radicles or rarely intra-ductal debris or stones whereas bile lakes and bile duct dilation typically not seen. Roux-en-Y loop obstruction should be considered, and a nuclear medicine scan can help to assess hepatic excretion as well as function. Treatment with antibiotics, preferably via intravenous route, needs to be considered. Recurrent cholangitis is an indication for listing for liver transplantation.

The presence of portal hypertension and development of varices on endoscopy during adolescence has been shown to increase the risk of requiring liver transplantation in adulthood, 7 and 8.5 fold, respectively. Assessment of presence of oesophageal and gastric varices is particularly important in young women who are considering pregnancy as well as during pregnancy to assess the risk of GI bleeding and decide on further management. Westbrook et al. [52]. found that a platelet count of <110 × 109/L in women with cirrhosis predicted the presence of varices during pregnancy [53]. In addition, evidence of more advanced liver disease, defined as Model of end stage liver disease (MELD) score of >6 and United Kingdom end stage liver disease score (UKELD) >46, increased maternal and foetal risk during pregnancy [52]. In BA, 58 live births in 40 females have been reported to date and both cholangitis and variceal bleeding were risk factors for developing complications during pregnancy [53,54].

Less is known about pubertal development and menstruation in young people with biliary atresia. It is known that advanced liver disease alters the physiology of the hypothalamic–pituitary–gonadal axis and disturbs oestrogen metabolism, affecting sexual function. Prevalence of amenorrhoea in women with advanced liver disease has been estimated between 30 and 71%; however, case series are small [55,56]. After liver transplantation, the majority with have a normal menstrual cycle within the first year after transplantation. 

More recently, fibroscan has been used to assess portal hypertension in children and young people with BA and liver stiffness > 24 kPa has been shown to be a good predictor of portal hypertension in older children with BA [57].

In addition to portal hypertension and cholangitis, serum bilirubin levels just above the upper limit of normal (>21 umol/L) at the age of 12 years have shown to predict the need for LT in adulthood. Interestingly, serum sodium and creatinine levels included in adult LT allocation models such as Model for End stage Liver Disease (MELD) and United Kingdom End stage Liver disease (UKELD) do not reflect the severity of liver disease in BA therefore these should be interpreted with caution in this setting [52]. A recent report from the Scientific Registry of Transplant Recipients (SRTR) found that out of 331 patients with biliary atresia listed for LT, including 114 adolescents (12–17 years) and 217 adults (>18 years), adults demonstrated increased risk of waiting list mortality compared with adolescents (10.9 higher risk on multivariate analysis) [58]. Additionally, in comparison to the adult cohort, adolescents had lower laboratory MELD/Paediatric End-stage Liver Disease (PELD) score at listing and at LT but demonstrated superior 5 year patient (98% vs. 84%) and graft (94% vs. 79.5%), respectively. 

Jain et al. suggested that MELD > 8.5 and UKELD > 47 predicted LT > 16 years in 397 BA patients with 84% and 79% sensitivity and 73% and 73% specificity, respectively [59]. Other predictive scores were evaluated including Mayo Primary Sclerosing Cholangitis risk score (MayoPSC) which includes markers of portal hypertension and synthetic function. MayoPSC revealed predictive accuracy for LT (AUROC 0.859), with a score of >0.87 predicting LT with 85% sensitivity and 82% specificity. MELD and MayoPSC at the age of 12 years as well as change in MELD, PELD and MayoPSC between 12 and 16 years, was associated with the need for LT. 

Adults with BA requiring LT also provide surgical challenges in particular related to the vascular system. In a review of 36 adults with BA undergoing LT, 57.6% was found to have significant enlargement of the splenic artery with 21% noted to have multiple splenic artery aneurysms [51]. Spontaneous visceral porto-systemic shunting (SPSS) was present in 72.7%. Survival was excellent with 10-year graft and patient survival exceeding 90%, highlighting the importance of careful donor selection and transplant surgical expertise in this condition.

What is not clear is whether the presence of SPSS is associated with the concept of covert or minimal hepatic encephalopathy, well described in adults with cirrhosis but less explored in children and adolescents. Considering that school functioning is lower compared to controls, with 2–48% of children requiring additional educational support further research initiatives should focus on cognitive function in patients with BA and its relation with disease severity in order to improve social outcomes [60].

## 6. Summary

Advances in both medicine and surgery have significantly improved the prognosis for infants born with biliary atresia. Although long-term survival with native liver occurs in the minority, survival rates post liver transplantation are high. Most young people born with BA are likely to survive into adulthood when they need to adapt to managing their own care within adult services. Key to this success is an integrated transition service between paediatric and adult care, supported by a multidisciplinary team to ensure good clinical and psychosocial outcomes. Adult hepatologists should be aware of the clinical management and complications of biliary atresia and the long-term consequences of liver transplantation in childhood and work closely with their paediatric colleagues to achieve a successful transition.

## Figures and Tables

**Figure 1 jcm-12-01594-f001:**
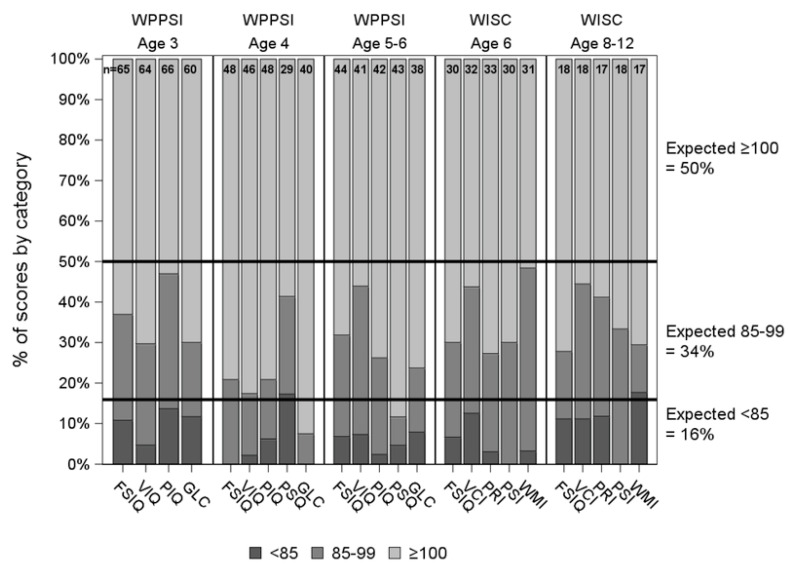
IQ scores in children with BA ages 3–12 years. Neurocognitive Score Distribution. Distribution of WPPSI-III and WISC-IV scores compared to population norms. FSIQ: Full Scale Intelligent Quotient; GLC: General Language Composite; PIQ: Performance Intelligent Quotient; VIQ: Verbal Intelligent Quotient; PSQ: Processing Speed Quotient; PRI: Perceptual Reasoning Index; PSI: Processing Speed Index; VCI: Verbal Comprehension Index; WMI: Working Memory Index [8].

**Figure 2 jcm-12-01594-f002:**
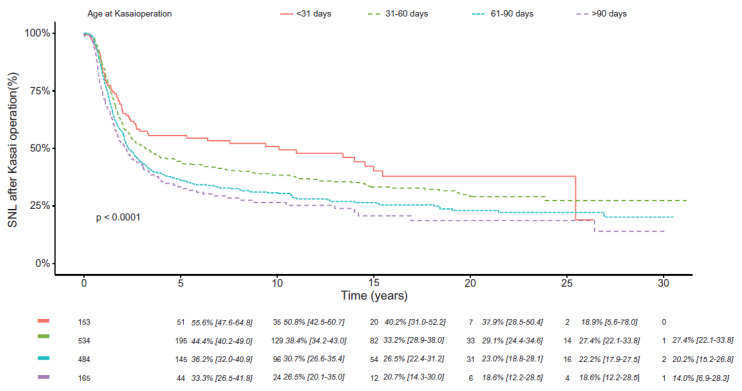
Survival with native liver according to age at Kasai operation [2]. At each point of follow-up: N patients alive with native liver, SNL after Kasai operation (CI 95%). SNL = survival with native liver [2].

**Figure 3 jcm-12-01594-f003:**
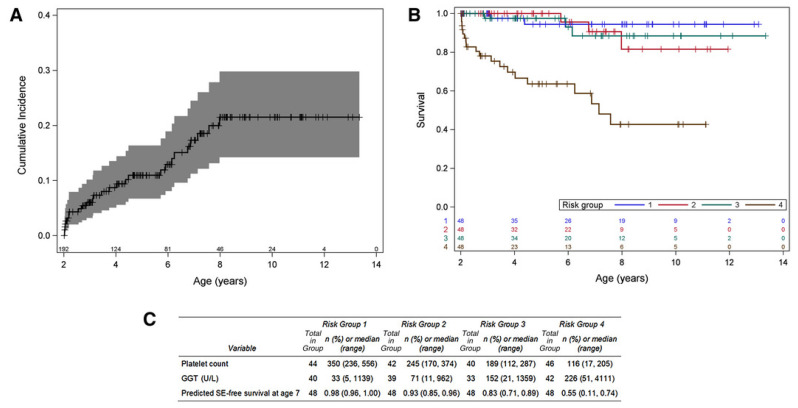
Incidence and risk for the SE-model (sentinel event) (**A**) Cumulative incidence of LTD among the 240 study participants. (**B**) Kaplan-Meier curves for LT-free survival stratified by quartile of risk score. Stratification of participants shows a high-risk group (group 4; brown) and a medium-risk group (group 3; green), with the remaining two quartiles showing a similar lower risk. (**C**) Risk factor distribution of participants in the analysis by quartiles of risk is provided. Venkat [18].

**Table 1 jcm-12-01594-t001:** Reported outcomes of BA worldwide in main registries [2].

	Europe							America		Asia	
	France (this study)	France (last 3 cohorts)	UK (16)	Switzerland (19)	Netherlands (17)	Netherlands (18)	Nordic countries (20)	Canada (21,22)	USA: BARC (9 Expert centers) (23)	Taiwan (15)	Japan (13)
Years	1986–2015	1997–2015	1999–2009	1994–2004	1977–1988	1987–2008	2005–2016	1985–2002	1997–2000	2004–2009/2010 *	1989–2015
Max follow-up	30 y	20 y	10 y	10 y	31 y	22 y	12 y	18 y	2 y	7 y	28 y
N patients	1428	951	443	48	104	231	154	349	104 (All after Kop)	197 (170 *)	3160
N Kasai op	1340 (94%)	895 (94%)	424 (96%)	43 (90%)	104 (100%)	214 (93%)	148 (96%)	312 (89%)	104 (-)	193 (98%)	3090 (98%)
Age at KOp, days	59 (6–199)	57 (6–199)	54 (7–209)	68 (30–126)	59 (25–222)	59 (20–210)	60 (4–165)	65 (6–200)	61	Term: 53 +/− 19	68
Median (range) or mean ± SD										Preterm: 72 +/− 28	
Documented clearance of jaundice after Kasai op	38%	39%	55%	39.5%	–	34%	64%	NA	NA	Term: 62% Preterm 37% (bilirubin < 34 µmol/L)	58% ^†^
SNL after K op											
At last follow-up	34%	41%	50%	40%	–	–	55% ^‡^	27%	56%	18 months SNL:	
10 y (CI 95%)	35% (34–36)	36% (34–38)	40% (34–46)	33% (26–40)	–	4 y: 46% (42–50)	45% (35–55) ^‡^	26% (20–32)	–	Term 73%	–
20 y (CI 95%)	26% (24–28)	26% (24–28)	–	–	27% §	–	–	–	–	Preterm: 50%	–
30 y (CI 95%)	22% (20–24)	–	–	–	–	–	–	–	–		49%
Death without LT	10%	7%	6%	8%	50%	–	8%	12%	4% (all after Kop)	–	–
N transplanted patients at last follow-up	793 (56%)	512 (54%)	194 (44%)	27 (56%)	26 (25%)	69 (32%)^‖^	64 (42%)	210 (60%)	42 (40%)	41 (20%)	1236 (39%)
Survival after LT					‖						
At last follow-up	84%	90%	93%	100%	73%	NA	86%	80%	88%	32 (78%)	1134 (92%)
5 y (CI 95%)	85% (84–86)	90% (89–91)	–	100%		4 y: 79% ^‖^	84% (74–94)	4 y: 83% (77–88)	–	–	
10 y (CI 95%)	84% (83–85)	90% (89–91)	–	–		–	84% (74–94)	80% (75–86)	–	–	
20 y (CI 95%)	80 (78–82)	89% (87–91)	–	–		–	–	–	–	–	
28 y	79 (77–81)	–	–	–		–	–	–	–	–	
Overall survival											
At last follow-up	81%	87%	91%	92%	43%	NA	87%	77%	91%	–	
5 y (CI 95%)	82% (81–83)	87% (86–88)	90% (88–93)	92% (87–96)	–	4 y: 73 (70–76) ^‖^	88% (83–94)	4 y: 77 (72–82)	–	–	–
10 y (CI 95%)	80% (79–81)	86% (85–87)	89% (86–93)	–	–	–	87% (81–93)	75% (70–80)	–	–	–
20 y (CI 95%)	78% (77–79)	85% (84–86)	–	–	–	–	–	–	–	–	89%
30 y (CI 95%)	76% (74–78)	–	–	–	43%	–	–	–	–	–	–
Population 2018 (million people)	68	66	8.6		17		27	36	326	24	127
Incidence of BA (CI 95%)	1/19,600 (1/18,200–1/21,100)	1/17,000 –	1/17,800 (1/13,900–1/24,800)		1/18,600	–	–	1/19,000 (1/17,800–1/20,300)	–	1/6600 (1/ 4200–1/7000)	1/13,500 (1/11, 300–1/17,000)
% of BASM	8.2%	11% (37)	8%		5%		12%	14%	11%	–	2%

BA = biliary atresia; BARC = Biliary Atresia Research Consortium; BASM = BA splenic malformation syndrome; KOp = Kasai operation; LT = liver transplantation; SNL = survival with native liver. * Survival analysis made on 170 patients with BA born between 2004 and 2009. ^†^ Value calculated from text and Figure 2 and Figure 4 of reference 13. ^‡^ Including 6 of 154 patients who did not undergo Kasai operation. ^§^ 20 y SNL: 1977–1982 20%; 1983–1988: 32%. ^‖^ Six patients who underwent LT abroad excluded from analysis.

**Table 2 jcm-12-01594-t002:** Predictors of long-term outcome of BA in survivors of the Kasai with a native liver up to 30 years post Kasai [2].

Univariate Analysis						
		SNL _ SE standard error (number of patients alive with native liver at follow-up)				
Prognostic factor	N patients	5-Y SNL	10-Y SNL	20-Y SNL	30-Y SNL	*p*
Anatomical pattern of the extra hepatic biliary remnant						<0.0001
Type 1	20	89.7% ± 6.9% (18)	84.1% ± 8.4% (16)	75.7% ± 11.2% (11)	75.7% ± 11.2% (11)	
Type 2	104	60.4% ± 5.1% (47)	51.6% ± 5.7% (26)	44.3% ± 6.3% (15)	36.9% ± 8.6% (6)	
Type 3	235	48.2% ± 3.4% (92)	42.9% ± 3.5% (68)	29.9% ± 3.8% (25)	29.9% ± 3.8% (25)	
Type 4	917	36.0% ± 1.7% (264)	30.7% ± 1.6% (184)	23.1% ± 1.8% (48)	18.1% ± 2.8% (11)	
Missing data	64					
BASM syndrome						<0.0001
No	1002	44.0% ± 1.6% (366)	38.5% ± 1.6% (258)	28.4% ± 1.8% (72)	24.7% ± 2.4% (19)	
Yes	118	20.8% ± 3.8% (20)	15.4% ± 3.7% (11)	15.4% ± 3.7% (11)	7.7% ± 5.7% (2)	
Missing data						
Age at Kasai operation						<0.0001
<31 days	153	55.6% ± 4.4% (60)	50.8% ± 4.6% (39)	37.9% ± 5.5% (17)	18.9% ± 13.7% (2)	
31–60 days	534	44.4% ± 2.2% (197)	38.4% ± 2.2% (134)	29.1% ± 2.6% (34)	27.4% ± 2.9% (17)	
61–90 days	484	36.2% ± 2.3% (146)	30.7% ± 2.2% (102)	23.0% ± 2.4% (35)	20.2% ± 2.9% (11)	
>90 days	165	33.3% ± 3.8% (45)	26.7% ± 3.7% (27)	18.6% ± 4.0% (10)	14.0% ± 5.0% (4)	
Missing data	4					
Multivariate analysis						
Prognostic factor		Hazard ratio		95% CI		*p*
Anatomical pattern of the extrahepatic biliary remnant						
Type 1		0.145		0.046–0.453		0.0009
Type 2		0.531		0.387–0.728		
Type 3		0.746		0.605–0.920		
Type 4		1				
BASM syndrome						
No		0.550		0.438–0.691		<0.0001
Yes		1				
Age at Kasai operation						
<31 days		0.538		0.388–0.745		0.0002
31–60 days		0.616		0.488–0.779		
61–90 days		0.766		0.604–0.971		
>90 days		1				

BASM = BA splenic malformation syndrome; CI = confidence interval; SE = standard error; SNL = survival with native liver.

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
