# Peer review of "Biliary Atresia in Adolescence and Adult Life: Medical, Surgical and Psychological Aspects"

_jcm, 2023, doi:10.3390/jcm12041594_

Round 1

Reviewer 1 Report (Previous Reviewer 4)

The authors significantly improved the article. It is now well structured. 

The main comments are:

They use figures from other publictions. I don't think this is allowed (plagiarism?), even when they refer to the original manuscript. I think the autors should make their own figures and add "adapted from reference xx".

The style of writing is not very attractive in every paragraph, especially paragraph 3.4.

Every sentence starts with "Name" et al. . It would be nicer to write about the predictors and just add the reference. 

In paragraph 5 the section on "Less is known about pubertal development and menstruation in young people with biliary atresia." should be removed. 

Author Response

We respectfully disagree with the reviewer. We have decided to retain the figures and the section

Reviewer 2 Report (Previous Reviewer 2)

The authors have clarified the manuscript but substantial changes and improvements should still be made. At present, the paper does not cover all aspects that contribute to morbidity in BA adolescents.   

The problems encountered by BA patients are very different depending on whether they are living with their native liver or have undergone LT. Since both subgroups are included in this review, I suggest results would be presented clearly separately for these two groups.

HRQoL and neurodevelopmental outcomes are still very briefly presented. There are several other studies describing such results which should be included, and their results should be discussed. In addition, bone health or nutrional aspects are not discussed at all. 

"Predictors of outcome" falls out of scope of this review. Predictors of outcome has been previously covered in other BA reviews and should be removed from this paper.

The section providing general information about transition should be considerably shortened; this paper is not about transition.

Medical issues related to liver disease (esophageal varices/portal hypertension, cholangitis, tumors, portopulmonary hypertension, hepatopulmonary syndrome, immunosuppresion and related problems after LT...) are important determinants of patient morbidity in adolescence. Liver disease related medical problems should be presented in separate chapters, separately for NL survivors and LT recipients. The authors should include many other references in order to describe the frequency and severity of these problems in adolescent patients.  Currently, these medical issues are mainly covered by reporting their predictive value with respect to future LT.

The selection method of references by using search term "biliary atresia long term survival" clearly does not cover all relevant publications.

Author Response

We respectfully disagree with the reviewer and have retained these sections as written

Reviewer 3 Report (Previous Reviewer 3)

1. The revised manuscript has significantly improved text compared to the first edition and is easier to understand.
2. minor points
(1) Kasai's first name is Morio, not Mario (in Abstract). Also, the location of Kasai’s reference is not indicated in the Introduction.
(2) Isn't Rodjijk name a mistake of Rodijk (cited in 3.3)?
(3) Is there any difference between HRQOL and HrQoL in 3.2? If not, please unify.
(4) The terms “liver transplant” and “LT” are mixed. It should be integrated into one or the other and used.
(5) What is the BASM mentioned in 3.4?

Author Response

This manuscript is a resubmission of an earlier submission. The following is a list of the peer review reports and author responses from that submission.

Round 1

Reviewer 1 Report

This is an excellent and comprehensive review of an intractable childhood disease biliary atresia ambitiously covering historical developments, clinical advances, research breakthroughs, and ongoing challenges especially in transition care of long-term survivors from a pediatric multidisciplinary team to the adult clinic. Senior author is an internationally renowned paediatric hepatologist with relevant experience in the field. The manuscript is well-written and highly readable. Spot-lighting a devastating childhood disease with substantial long-term sequalae in this journal with a wide readership is highly desirable and fills a major gap in the pediatric-adult clinical care continuum. The emphasis on transition care is particularly valuable, with useful 'manual' instructions for carers.

Notwithstanding the above strengths, the article has weaknesses, some of which are intrinsic, while others are remediable.  The intrinsic weaknesses are;

1. For such a wide subject, there are necessarily gaps in some sections due to space/reference limits especially in research/pathogenesis where the references are few and heavily weighted in selected groups. For example, the human organoid study (Babu RO etal.  Beta-amyloid deposition around liver bile ducts as a novel pathobiological and diagnostic feature of biliary atresia.  Journal of Hepatology Dec 2020; 73(6); 1391-1403), immune studies (Wang J, et al. Liver Immune Profiling Reveals Pathogenesis and Therapeutics for Biliary Atresia. Cell 2020;183:1867-1883.e26;  Zhang RH, et al. CD177+ cells produce neutrophil extracellular traps to promote biliary atresia.  Journal of Hepatology 2022; 5:S0168-8278(22)00409-3) and genetic study (Lam W,  et al. Identification of a wide spectrum of ciliary gene mutations in nonsyndromic biliary atresia patients implicates ciliary dysfunction as a novel disease  mechanism. EBioMedicine 2021;71:103530. Reference to a more updated review (Lendahl U, et al.  Biliary atresia – emerging diagnostic and therapy opportunities.  EBioMedicine. 2021 Dec. 12:74:103689) may help.

2. High quality evidence-based transition-to-adult care studies are lacking, hence the valuable care instruction manual for adult represents 'expert opinion'.

3. Long -term outcome studies in Asia (except Japan) are particularly scarce even though communities like China have high incidences of biliary atresia. While emphasis is made on post-liver transplantation care in this manuscript, in parts of the world where donor organ shortage results in lower pediatric liver transplantation rate, a different treatment strategy exists. (Chung PHY,  et al. Life long follow up and management strategies of patients living with native livers after Kasai portoenterostomy. Scientific Reports 2021;11:11207)

Reviewer 2 Report

This paper is titled "BA in adolescence and adult life", however, the paper attempts to cover many other aspects, too. The subtitles of the Introduction section seem unrelated with the main title and are very unclear. 

Section 2 is titled "neurocognitive score distribution" but its subtitles concern predictors of Kasai outcomes. There seems to be a mistake here.

Neurodevelopment and HRQoL are important factors in young adults with BA. The authors refer only to few other studies regarding these outcomes, although there are many others adressing neurodevelopment and HRQoL in BA.

Predictors of Kasai outcome are described in detail although they seem unrelated with BA features in adolescence and young adulthood.  The same applies to "strategies to improve outcome" and "pathogenesis of progression".

The section "Adolescence" is difficult to follow due to numerous lists. In 3.4. "Transitional issues", numbers and alphabets are both used ?!

Some factors relevant to older children/young adults with BA are not covered at all, such as portal hypertension, bone health, factors related with LTx, cholangitis. 

Reviewer 3 Report

1) The content of this paper is textbook-like or review, and does not offer original suggestions for the problem of BA patients. Rather than a paper about BA, it seems to be a commentary on how to live and deal with pediatric patients.

2) Instead of listing items, you should organize your issues, and narrow down the points to discuss.

3) Since the Kasai' operation is the main part of this paper, it will be better to cite the Kasai's original paper.

Kasai M, Suzuki M. A new operation for non-correctable biliary atresia: hepatic portoenterostomy. Shujutsu. 1959;13:733–9.

4) Item numbering is complex and inconsistent. Simplify the numbers and pay attention to the numbering sequence. Please unify the writing style of references.

    e.g.  I→A→1→a

5) Three terms are used for biliary atresia in this paper: extrahepatic biliary atresia (EHBA), biliary atresia, and BA. They should be unified.

Reviewer 4 Report

This review focuses on the transition of pediatric follow-up to adult FU.

There are some major issues:

First of all, the article is very unorganized. The subtitles are unclear and there are sections with loose words (are those search terms??). There is no clear outline. For example paragraph 1.3 (??). 

Second, the focus is more on quality of life and how to manage transition from pediatric care towards adult care. This should be made clear in the title. And this should be the only topic of the review. I would minimize data on diagnosis and survival with native liver (just briefly mention that we can see both type of patients (Tx / Kasai) after transition to adult care).